# DoA Estimation for FMCW Radar by 3D-CNN

**DOI:** 10.3390/s21165319

**Published:** 2021-08-06

**Authors:** Tzu-Hsien Sang, Feng-Tsun Chien, Chia-Chih Chang, Kuan-Yu Tseng, Bo-Sheng Wang, Jiun-In Guo

**Affiliations:** 1Institute of Electronics, National Yang Ming Chiao Tung University, Hsin-Chu 300, Taiwan; ftchien@mail.nctu.edu.tw (F.-T.C.); john86328@gmail.com (C.-C.C.); p0905550162@gmail.com (K.-Y.T.); jiguoccu@gmail.com (J.-I.G.); 2Institute of Electronics, National Chiao Tung University, Hsin-Chu 300, Taiwan; shaw324love@gmail.com

**Keywords:** FMCW radar, deep learning, three-dimension convolution network, direction-of-arrival estimation

## Abstract

A method of direction-of-arrival (DoA) estimation for FMCW (Frequency Modulated Continuous Wave) radar is presented. In addition to MUSIC, which is the popular high-resolution DoA estimation algorithm, deep learning has recently emerged as a very promising alternative. It is proposed in this paper to use a 3D convolutional neural network (CNN) for DoA estimation. The 3D-CNN extracts from the radar data cube spectrum features of the region of interest (RoI) centered on the potential positions of the targets, thereby capturing the spectrum phase shift information, which corresponds to DoA, along the antenna axis. Finally, the results of simulations and experiments are provided to demonstrate the superior performance, as well as the limitations, of the proposed 3D-CNN.

## 1. Introduction

Radar is considered to be one of the key sensor technologies for enabling autonomous driving [1], and mmW radar in particular has become very popular because of its advantages, especially compared to LiDAR (light detection and ranging), such as small size, low transmit power, excellent range resolution, low clutter sensitivity, as well as the relatively low cost with competitive performance under almost all weather conditions [2,3]. The most commonly used waveform in low-cost mmW radar today is the frequency modulated continuous wave (FMCW). By utilizing the large bandwidth available and the high carrier frequencies in automotive radar bands, good resolution can be achieved in range and speed estimation. However, the angle resolution is limited by the number of antennas of the phased-array. Since the automotive radar sensor is a mass-produced product, cost is an important consideration, so the number of receiving antennas is usually low, in the tens [4,5].

High-resolution DoA estimation has recently become very relevant in the current filed of autonomous driving and advanced driver assistant system (ADAS). Conventional high-resolution DoA estimation algorithms, such as the popular MUSIC algorithm and its variants [6,7,8,9], are based on sub-space methods. In the application of automotive radars, usually only a few measurement snapshots can be used. However, the sub-space methods can only achieve the best performance when enough snapshots are available. Because they are based on presumed signal models, the performance of these algorithms degrade fast when the signal-to-noise ratio (SNR) becomes low. In the scenarios typical in vehicular transportation, accurate DoA estimation becomes challenging [10,11]. In addition, when the number of detected targets is unknown, and multiple targets have different radar cross-section (RCS), their performance may also be affected. The correct detection of the number of targets in the receiver signal is also an important issue in array signal processing [12].

The aim of this paper is to present a 3D-CNN for high-resolution DoA estimation for FMCW radar. Up to two targets will be distinguished and their DoAs be estimated. Region of interest (RoI) is used to manage cases with more targets. Two formats of data are used in the process: the raw radar data cube and the statistical average in the form of covariance matrices. Simulation and experiment results are provided to demonstrate the superior performance of the proposed method over conventional methods such MUSIC as well as emerging deep learning methods [13,14,15,16,17,18,19,20]. It has been claimed, based on simulation results, that neural networks (NNs) can outperform MUSIC when SNRs are relatively low; however, no convincing experimental results have been shown to support the claims [14,15,16,17,18,19,20], and the angular differences between targets are not small enough to show the supremacy of deep-learning-based methods [20].

In this paper, experimental results in addition to simulation results are presented to demonstrate the superior performance of the proposed 3D-CNN. To overcome the issue of lacking the diversity of range in the training data set, data augmentation techniques are also developed. For the critical cases of two targets with small angular differences (≤4°), our 3D-CNN shows very promising results of resolving the angles. Overall, compared to DNN (Deep NN) [19], the 3D-CNN achieves 3% improvement in accuracy for the two-target scenario over a large range of SNRs in simulations, and 2% improvement in experiments. In addition, it is relatively easy to extend the proposed approach for processing the data cube to obtain the target’s other parameters in the future. 

## 2. Materials and Methods

The task of estimating DoAs can be divided mainly into two stages, the first being signal preprocessing and the second being DoA estimation by CNN. The two stages will be described in the following two subsections. Furthermore, details of data generation and data augmentation in the simulation/experiment are also provided.

### 2.1. Signal Model and Signal Preprocessing

In this paper the conventional model of plane waves impinging a uniform linear array (ULA) is adopted [21]. Suppose that plane waves reflected from N targets with complex amplitudes $S_{n}\left(t\right)$ and wavelength λ come from directions $\theta_{n},\;n=1,\cdots ,\;N$. The ULA contains M antennas spaced by d. After demodulation, the received signal of the ULA at snapshot time t can be formulated as:(1)x(t)=As(t)+n(t),
where x(t), s(t), and n(t) are:(2)$$x\left(t\right)={\left[x_{1}\left(t\right),\ldots ,x_{M}\left(t\right)\right]}^{T},s\left(t\right)={\left[S_{1}\left(t\right),\ldots ,S_{N}\left(t\right)\right]}^{T},n\left(t\right)={\left[n_{1}\left(t\right),\ldots ,n_{M}\left(t\right)\right]}^{T}.$$

In the above equations, s(t) (an N×1 vector) denotes the reflected signals coming from the far-field DoAs $\theta_{1},\ldots ,\theta_{N}$, and n(t) is an M×1 vector of independent additive white Gaussian noise (AWGN) with zero mean and variance of $\sigma^{2}$.

Define an *M* × *N* matrix A whose columns are steering vectors toward N DoAs:(3)$$A=\left[a\left(\theta_{1}\right),\cdots ,a\left(\theta_{N}\right)\right],$$
with,
(4)$$a\left(\theta_{n}\right)={\left[1,e^{-j\eta_{n}},\;e^{-j2\eta_{n}},\cdots ,\;e^{-j\left(M-1\right)\eta_{n}}\right]}^{T},$$
where $a(\theta_{n})$ is the steering vector and $\eta_{n}=2\pi d\sin\left(\theta_{n}\right)/\lambda$ is the phase difference between adjacent elements. The demodulated signals contain enough information to estimate the DoAs.

Many DoA estimation methods use the M×M covariance matrix of x(t):(5)$$R_{x}=E\left[x\left(t\right)x^{H}\left(t\right)\right]=AR_{s}A^{H}+R_{N},$$
where ${\left[.\right]}^{H}$ and E[.] denotes the conjugate transpose and expectation operator. $R_{s}$ and $R_{N}$ are the K×K signal and noise covariance matrices, respectively. It is assumed that noise components are AWGN and the covariance matrix is:(6)$$R_{N}=\sigma^{2}I,$$
where I is the identity matrix and $\sigma^{2}$ is the noise power. In practice, $R_{x}$ is unknown, and the number of sampled signal vectors is limited. The maximum likelihood estimation of the spatial covariance matrix can be obtained from L snapshots by:(7)$${\hat{R}}_{x}=\frac{1}{L}\sum_{l=1}^{L}x\left(l\right)x{\left(l\right)}^{H}.$$

The demodulated signal vectors contain sufficient information for DoA estimation. However, the size of data vectors can be huge that the training of NNs is greatly complicated. In radar applications, the signals reflected from multiple targets arrive at the receiver at different times. It is desirable to use this fact to first do a coarse separation of target signals such that the training can be simplified. Therefore, RoI will be first established to limit the coverage of the NN, and target signals with very close ranges and DoAs in the range-azimuth spectrogram will be grouped into the same RoI for further processing.

The FMCW radar is chosen to further build the signal model due to its popularity in vehicular applications [1,2] especially the signal framework of TI mmW radar AWR1843BOOST + DCA100EVM is adopted [22]. The process of generating a data cube [23] and selecting RoI in the proposed scheme is summarized in Figure 1 and Figure 2. First, multiple snapshots are collected and each is transformed by DFT to generate range information. Afterwards all snapshots are stacked along the antenna and chirp axis to form a data cube for further processing. The selection of RoI is made by first detecting possible targets on the range axis by using the cell-averaging constant false alarm rate (CA-CFAR) principle [24,25,26]. Then the J samples centered at the target will be claimed as RoI. If needed, a coarse DoA/speed interval can also be obtained by conventional DoA/speed estimation methods such as angle/speed DFT along the antenna/chirp axis [23]. In this paper, J=65 samples along the range axis and the angular interval from −10° to 10° are used for delineating the RoI. Signals of interest that are not in the interval [−10°, 10°] can be first rotated into the interval by the same technique described by (11) and (12) in the subsection on data augmentation; the DoA estimation can then be carried out and the results be rotated back to the correct directions. The three-dimensional RoI of the data cube will be further cropped and manipulated as the input of the deep learning framework.

The data cube is complex-valued. The NN will take the absolute value, the real and the imaginary part of the data as the input for DoA estimation. A similar data format has been found to be effective for training neural networks [27]. In addition to the RoI of the data cube, the covariance matrix is also used as the input for determining the number of targets. The details will be given in the next subsection. In order to reduce the impact from noise outside the RoI, the covariance matrix is estimated with only the signal within the RoI. As shown in Figure 3, the data are first filtered by a band-pass filter fitted to the RoI, then the maximum likelihood estimation ${\hat{R}}_{x}$ in (7) is obtained. Since ${\hat{R}}_{x}$ is Hermitian, the diagonal terms and upper triangular part of it contains all the information. Let ${\hat{R}}_{x}$ be first normalized so that its real and imaginary parts are within [−1,1] and then be morphed into a vector $R\in R^{M\left(M-1\right)\times 1}$ for the input to a NN:(8)$$R=\left[Re({\hat{R}}_{12}\right),\;\;Im\left({\hat{R}}_{12}\right),\cdots ,\;Im({\hat{R}}_{1M}),\cdots ,\;Re({\hat{R}}_{23}),\cdots ,\;Im({\hat{R}}_{\left(M-1\right)M\;})]$$

### 2.2. DoA Estimation by CNN

Deep learning has breached into many application areas where conventional signal processing techniques are facing difficulties; among numerous neural network models, CNNs are known to be very capable of extracting features from matrix- or tensor-formed data [28,29,30,31]. Its architecture typically consists of one or more convolution layers followed by fully-connected networks leading to the output. In the proposed scheme, because a 3-D data cube is processed to make high-precision DoA estimation, a 3D-CNN [30] is deployed. However, before the 3D-CNN, a simple fully-connected NN is first used for determining the number of targets. The overall signal flow together with the signal preprocessing is summarized in Figure 4, and more details are itemized in the following.
Signal preprocessing and the selection of RoI have been described in the previous subsection.An independent small-size fully connected NN is used to estimate the number of targets. The signal’s statistical average is found to be more robust to provide coarse information, such as the number of targets; while the data cube contains subtle information without being averaged out so that it enables high-resolution DoA estimation. The input of the simple fully-connected NN to estimate the number of targets is an M×M covariance matrix (in our case, M=8), and the covariance matrix is obtained by (8). The estimated number (0, 1, or 2) will then be used to choose the one-target or two-target 3D-CNN for estimating the DoAs.The 3D-CNN does DoA estimation by treating it as a classification task [32]. There are 21 classes (corresponding to DoAs −10°,−9°, ⋯,10° in the RoI) to be classified. The angular resolution is set to 1° for two reasons. First, 1° is commonly regarded as a minimum requirement for applications in ADAS [1]. Second, higher angular resolution will lead to a larger 3D-CNN and increase the risk of facing more difficulties in training. In Figure 4, the highlighted RoI is the input to the 3D-CNN and it is a J×M×C×3 tensor (in our case, J=65, M=8, C=4, which stand for number of fast-time samples, of antennas, and of chirps respectively). The 3 is the number of data formats that include the absolute value, the real part, and imaginary part; using the arrangement it is shown to yield better performance than only using the real and imaginary parts [27]. The first convolutional layer in the 3-D CNN (Conv1) uses 30 × 5 × 3 kernels with the stride step of 1 × 1 × 1, and the number of channels is 256. The same number of channels are passed to the second convolutional layer (Conv2), which uses 2 × 2 × 2 kernels, and a max pooling (MP) layer follows. The output of feature extraction is then flattened and connected into the form of a matrix, and a four-layer fully connected layer is used for its classification. The loss function used for the classification task is binary cross entropy in the Pytorch environment [33].The parameters for the fully-connected NN and 3D-CNN are listed in Table 1 and Table 2. The term FC stands for fully connected and BN for batch normalization [34]. In the training of this fully-connect NN and the 3D-CNN, the popular optimization tool ADAM and the drop-out technique are used [35,36].

### 2.3. Data Generation and Data Augmentation

Both simulation data and experiment data are acquired for training the NNs and evaluating the proposed DoA estimation scheme. The simulation data are primarily for evaluating and comparing the performance of DoA estimation with the Cramer-Rao Bound (CRB) [37] and existing algorithms for a wide range of SNRs. The experimental data, on the other hand, are primarily for demonstrating the applicability of the proposed DoA estimation scheme.

Simulation data are generated by the MATLAB Phased Array tool box [38] with the parameters delineated below. To simplify the performance analysis, the number of targets is set to be 0, 1 or 2. The radar is an FMCW one with eight receiving antennas. The resonance frequency is 77 GHz and one snapshot received at each antenna has L=256 sample points. The DoA of each incident plane wave ranges from −10° to 10°, and signals under various SNRs ranging from −10 to 15 dB are generated. Detailed simulation parameters are summarized in Table 3. In total, 63,000 snapshots are generated for training and 2.1 k for each testing in different SNRs. Note that all the pairs of DoA ([−10°, −9°], [−10°, −8°], …, [9°, 10°]) are considered in both training and testing.

In simulation, the model is first trained by the first training data set with SNRs in the range from 15 to 0 dB; then the model is further strengthened by the second training data set with SNRs from 15 to −5 dB. Number of the both training data is 63 k. The reason for doing the two-stage training is to make sure the model first learns the correct signal features with relatively high-SNR data then make the model more robust by training it with more noisy data.

Experiments with metal corner reflectors mounted on tripods on a baseball field (Figure 5) are conducted to collect data. The TI mmW radar AWR1843BOOST+DCA100EVM is used (Figure 6); two Tx and four Rx antennas are used to constitute a virtual 8-antenna receiving array. Detailed experiment parameters are summarized in Table 4. Note that the sampling frequency of the device in the ultra-short range mode is 6250 kHz to accommodate the bandwidth of the demodulated signal (not the mmW signal). More comprehensive information about TI hardware can be found in [39,40]. Note that some radar parameters used in simulation and experiments are not exactly the same; this is due to the limitations of both the MATLAB Phased Array tool box and the TI radar platform. The training data set for experiment includes both 21 k raw snapshots (directly read from the TI radar) and 28 k snapshots with data augmentation, and about 7 k raw snapshots are reserved for testing. Details and methods of data augmentation are provided below. The reason of using augmented data is to increase the robustness of our trained model.

In addition to the data collected in the field experiment, data augmentation is deployed to increase the quantity and enhance the range diversity of the training data set. We suspect that the lack of reporting high-resolution results with using NNs on experiment data in literature might be just due to the robustness issue caused by insufficient range diversity. The experiments are conducted with the targets placed only at certain ranges. Nonetheless, real-world targets come with random ranges. Therefore, it is essential to expose the NNs to targets with a variety of ranges. One way to do this without trying to place targets at a multitude of ranges is implementing data augmentation. Note that the demodulated FMCW signal along the fast-time axis (in a data cube shown in Figure 2) exhibits a beat frequency corresponding to the target’s range [23]:(9)$$p\left(t\right)=e^{j2\pi \left[\left(\frac{2BR}{cT}\right)t+\frac{2f_{c}R}{c}\right]}$$
where p(t) is the demodulated FMCW chirp signal, R is the target’s range, $f_{c}$ is the carrier frequency, B is the bandwidth, T is the chirp duration, and c is the speed of light. Therefore, it only takes a proper frequency term to change the target’s range. As a result, we only need to multiply a phase term to signals along the fast-time axis to create virtual targets at the range R+ΔR, where ΔR is randomly chosen from −0.3 to 0.3 m:(10)$$p_{\text{range\_aug}}\left(t\right)=e^{j2\pi \left[\left(\frac{2B\Delta R}{cT}\right)t+\frac{2f_{c}\Delta R}{c}\right]}\times p\left(t\right).$$

From each angle set, there are several snapshots chosen to be augmented, and 220 random ΔR’s are generated to augment the size of data set to about 28 k.

It is also possible to augment data with angular rotations, if the angular placement of targets is not diverse enough, although not used in this paper. Let m and q denote the antenna and target index respectively (there are M antennas and Q targets in total), then the demodulated signal with multiple antennas can be expressed as [23]:(11)$$p\left(t,l\right)=\overset{Q}{\underset{q=1}{\sum }}\;e^{j2\pi \left[\left(\frac{2BR}{cT}\right)t+\frac{2f_{c}R}{c}+\frac{f_{c}d\sin\varphi_{q}}{c}l\right]}.$$

As can be seen, the frequency $\frac{f_{c}d\sin\varphi_{q}}{c}$ on the antenna axis contains the information of DoA $\varphi_{q}$. Augmented data with new DoAs can be obtained by modulating p(t,l) with a shifting frequency $\omega_{\mathrm{aug}}$:(12)$$p_{\text{doa\_aug}}\left(t,l\right)=e^{j\omega_{\mathrm{aug}}l}\times p\left(t,l\right).$$

## 3. Results

In this section, the simulation results of evaluating and comparing the performance of the proposed DoA estimation method with the Cramer–Rao bound (CRB) [37] and the DNN (Deep NN) in [19], which has the state-of-the-art performance among deep-learning-based method methods, is first presented. Then the experiment results with up to two corner reflectors are presented to demonstrate the potential of the proposed method in applying to real-world situations. Note that the proposed DoA estimation method consists of mainly two steps: the first is to determine with a fully-connected NN the number of targets in the RoI, while the second is to estimate the targets’ DoAs with a 3D-CNN designed either for one target or for two targets. Therefore, for both simulation and experiment, the accuracy of estimating the number of targets is first characterized, followed by the results on DoA estimation.

### 3.1. Simulation Results

Figure 7 shows the accuracy of determining the number of targets. As can be seen, the accuracy reaches 100% once the SNR hits 10 dB, and it almost maintains 95% even when the SNR is as low as 0 dB. The highly effective fully-connected NN provides a very ensuring condition for the 3D-CNN to work on DoA estimation.

Figure 8 presents the performance of DoA estimation for the case of one target. The root-mean-square error (RMSE) of the proposed method, MUSIC, the DNN in [19], and the CRB are compared. It is clear that both NN methods beat MUSIC and stay close to the CRB across a wide range of SNRs. The proposed 3D-CNN and the DNN in [19] have similar performance and both perform better than MUSIC in most SNRs. The advantage of the 3D-CNN becomes evident in the case of two targets, as will be shown next.

Note that for the two-target scenario, we could not find a proper definition of RMSE when the number of targets is estimated wrong; instead, the accuracy of DoA estimation, which is defined as the success rate of obtaining the correct target number and DoAs (with a ±1° tolerance of estimation error), is used for performance comparison. Four methods are compared in Figure 9. The original DNN proposed in [19] has about 0.135 million parameters, e.g., it is a very small DNN, especially compared to our 3D-CNN. It is no surprise that 3D-CNN has a clear advantage over the whole range of SNRs. However, the performance of the DNN improves when its size is increased. Therefore, we increase its size until the performance tops (after that large DNNs perform poorer due to difficulties in training), and the result is a large DNN with around 11 million parameters. As can been from Figure 9, the 3D-CNN still has the best overall performance, even though the large DNN has diminished the performance gap to 3 dB. It is also worth noting that at SNR = −10 dB, the 3D-CNN performs worse because the training data set does not contain data from this SNR. This shows that 3D-CNN is somewhat less robust than the large DNN.

### 3.2. Experimental Results

One-target and two-target experiments are conducted with the experimental set-up presented in Section 2.3. In the one-target scenario, all the three methods (3D-CNN, DNN, and MUSIC) estimate DoA correctly with 1° precision. Figure 10 shows the comparison of three NNs in terms of the accuracy of DoA estimation. The figure gives three sets of accuracy, i.e., with the tolerable angular deviation of 0°, 2° and 4°. Notice that MUSIC has been excluded from the comparison; it is because that MUSIC often fails to identify two DoAs and is not qualified for the competition. In the two-target scenario, with the tolerable angular deviation of 0°, the accuracy of 3D-CNN reaches 95.7%; while for the large DNN and DNN, the accuracy results are 93.7% and 89.1%, respectively. When larger angular deviation is allowed, however, the accuracy of large DNN again goes to the top. This again shows that DNN has somewhat more robust performance.

The 3D-CNN not only has better accuracy when zero angular deviation is allowed, it also excels in the critical cases where the angular difference between two targets is small. When the angular difference is small (≤4°), the 3D-CNN has an accuracy of 97.8%, while the DNN and the large DNN have 88.6% and 95.6%, respectively.

In summary, the 3D-CNN has the best performance, especially in the cases where the angular difference is small and the conventional method such as MUSIC faces most difficulties. On the other hand, the slight decline in its performance when facing low-SNR data that is not experienced in the training phase shows that its performance, as in many data-driven methods, depends greatly on the quality and quantity of the training data set available.

## 4. Discussion

DoA estimation is a classical problem for radar data processing. High-resolution methods such as MUSIC have been available for decades. Novel methods based on different types of NNs are widely regarded as the more promising approach for the old problem. In this paper we have demonstrated, as illustrated by Figure 6, Figure 7, Figure 8 and Figure 9, that two methods (3D-CNN and DNN) can consistently beat MUSIC with a large margin under both simulated and experiment scenarios. The 3D-CNN also has a better overall performance than the DNN, and it shows the potential of working with the RoI approach to generate high-resolution radar point clouds.

It may be argued that the better performance of 3D-CNN is due to the fact that its input data-format (3-D tensor) retains more information from the raw radar data than the DNN’s input (covariance matrix) does. On the other hand, the stability of data benefitting from the averaging operation in estimating the covariance matrix gives a slight advantage to the DNN, as demonstrated by the slightly more robust performance of DNN under some conditions.

Another advantage of 3D-CNN is its better potential in future development. For instance, its input data contains the information along the slow time; therefore it is possible to train the same network to estimate target velocity, i.e., using the same 3D-CNN to output target’s range, velocity and DoA simultaneously is possible.

Finally, not only does the input data format play an essential role in model learning, data augmentation is essential too. In acquiring the training data set, it is very difficult to conduct experiments covering comprehensively the range of targets. The neural networks, whether 3D-CNN or DNN, would perform poorly if the range information in the training set is not diverse enough. Equations (9)–(12) offers a way to cover range and DoA diversity in doing data augmentation for preparing a comprehensive training data set.

## 5. Conclusions

In this paper, an NN-based approach is proposed to estimate DoA for FMCW radar. The radar data cube is first pre-processed, based on the RoI of potential targets, to form the input for the NN. A simple fully-connected NN then estimates the number of targets before a 3D-CNN proceeds to estimate the DoAs of the targets. In order to overcome the lack of range diversity in the training data set, a data augmentation method is also developed. The results of simulations and experiments show that the 3D-CNN can effectively resolve the angular difference between two targets with a resolution up to 1° when the conventional high-resolution methods such as MUSIC fail due to low SNRs.

## Figures and Tables

**Figure 1 sensors-21-05319-f001:**
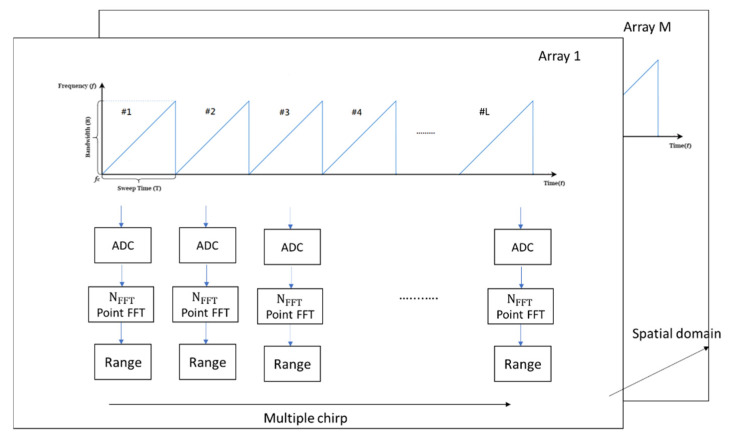
Generation of a data cube.

**Figure 2 sensors-21-05319-f002:**
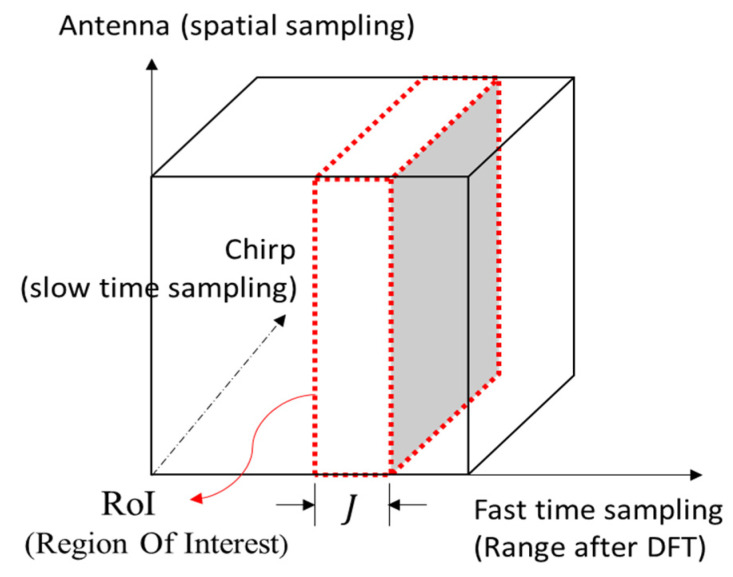
RoI selection in a data cube.

**Figure 3 sensors-21-05319-f003:**
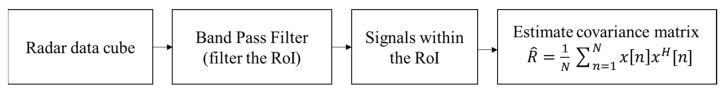
The covariance matrix of RoI.

**Figure 4 sensors-21-05319-f004:**
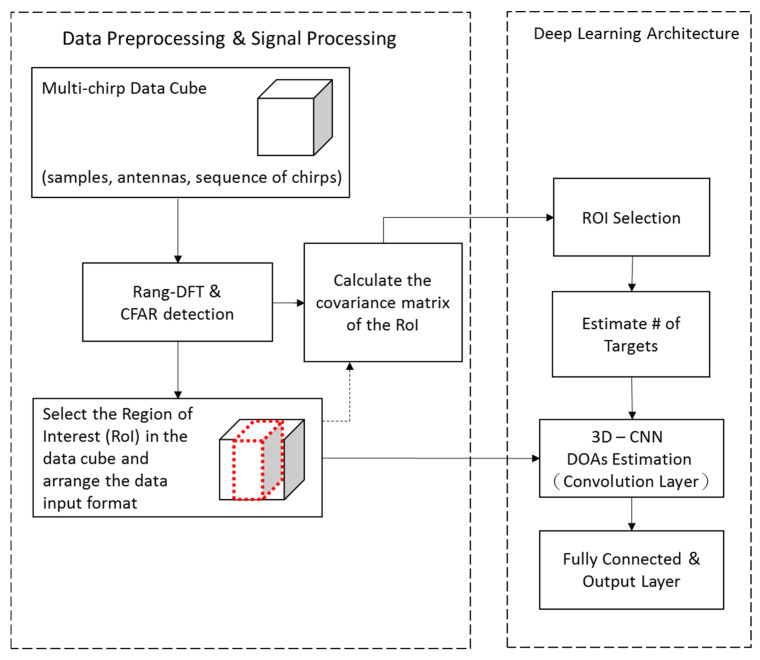
Signal processing and deep learning architecture.

**Figure 5 sensors-21-05319-f005:**
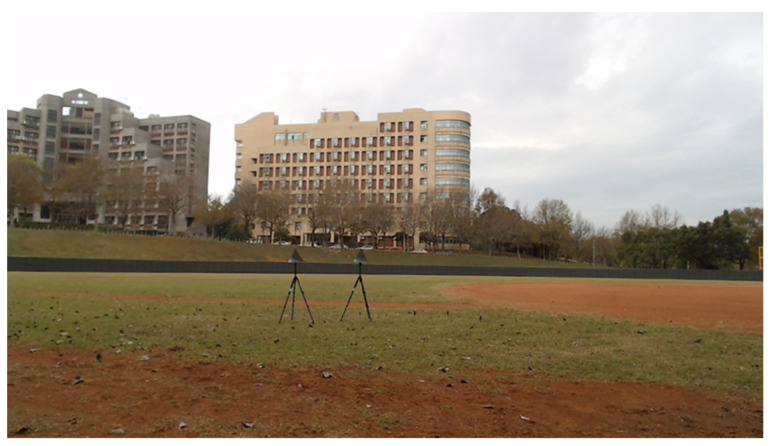
Experiments with two metal corner reflectors (targets).

**Figure 6 sensors-21-05319-f006:**
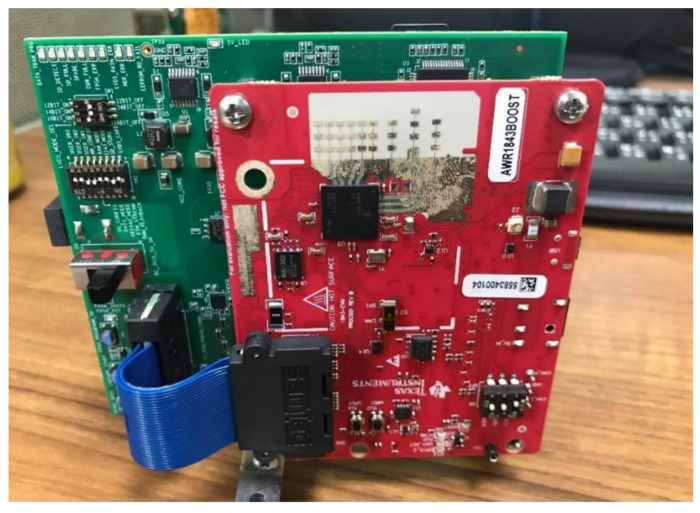
The photo of the actual radar used in experiments (AWR1843 and DCA1000EVM).

**Figure 7 sensors-21-05319-f007:**
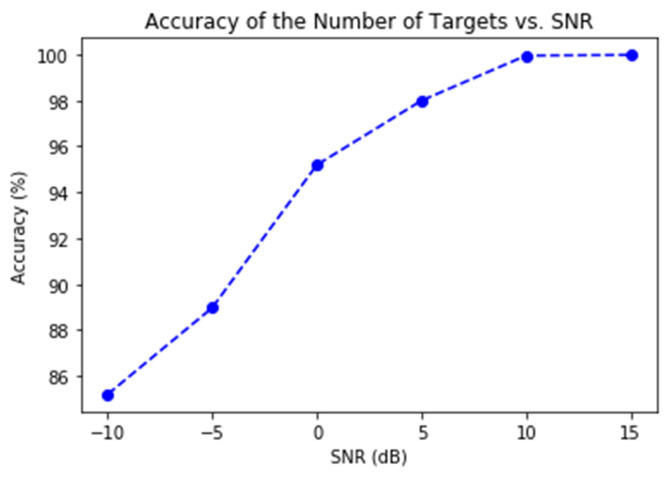
The accuracy of determining the number of targets vs. SNR.

**Figure 8 sensors-21-05319-f008:**
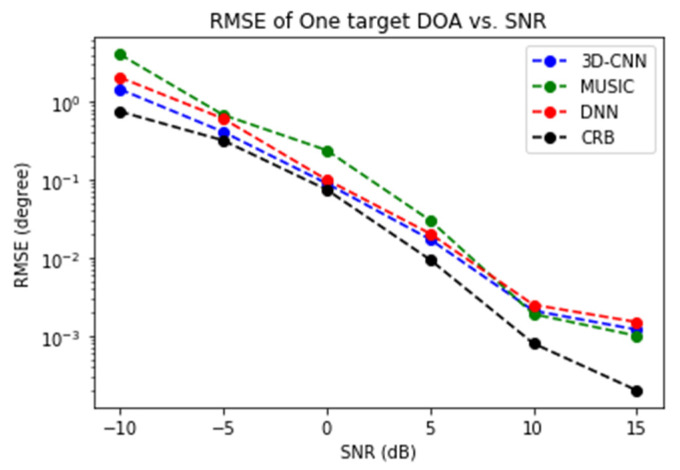
The RMSE of DoA estimation vs. SNR.

**Figure 9 sensors-21-05319-f009:**
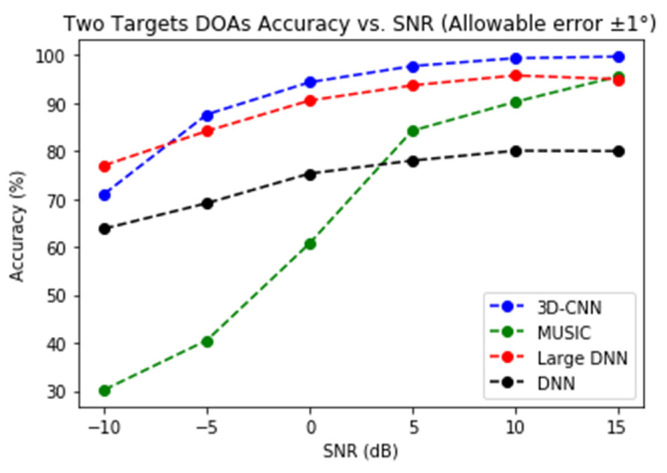
Accuracy of two-target DoA estimation, with an allowable error of ±1°.

**Figure 10 sensors-21-05319-f010:**
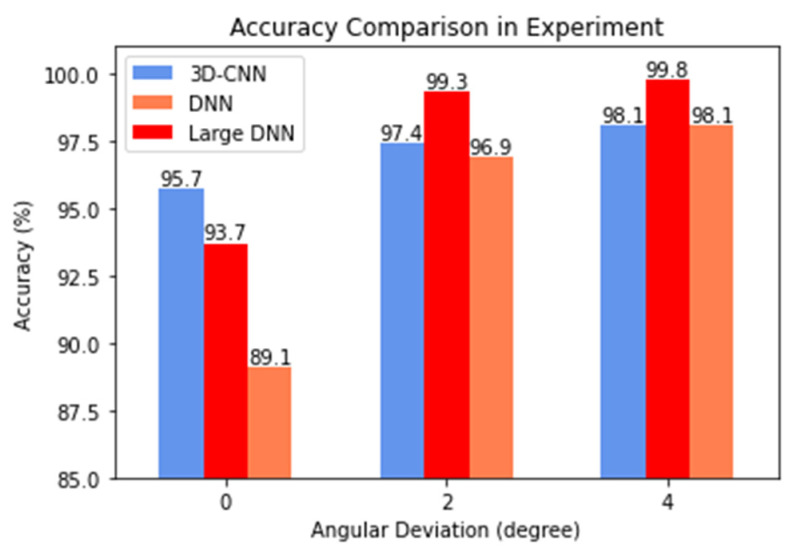
Accuracy of two-target DoA estimation vs. angular deviation.

**Table 1 sensors-21-05319-t001:** Fully-connected NN parameters.

Layer	No. of Filter	Activation	Layer	No. of Filter	Activation
Input	56		FC3 + BN	1024	ReLu
FC1 + BN	1024	ReLu	FC4 + BN	1024	ReLu
FC2 + BN	1024	ReLu	FC5 + BN	3	Sigmoid

**Table 2 sensors-21-05319-t002:** 3D-CNN parameters.

Layer	Filter	No. of Filter	Activation	Output
RoI spectrum				65 × 8 × 4 × 3
Conv1	(30, 5, 3)	256	ReLu	
Conv2	(2, 2, 2)	256	ReLu	
MaxPool 1	(2, 2, 2)			
FC			ReLu	500
FC			ReLu	200
FC			ReLu	100
FC			Sigmoid	21

**Table 3 sensors-21-05319-t003:** Simulation parameters (FMCW—Frequency Modulated Continuous Wave).

Parameters	Value	Parameters	Value
Transmit signal	FMCW	Target Number	0, 1 or 2
Carrier Frequency $f_{c}$	77 GHz	Target Range	70 m to 90 m
Frequency Slope B	3.476 THz/s	Target Angle	−10° to 10°
Bandwidth	0.15 GHz	Samples per chirp	256
Tx/Rx Antenna	1/8	SNR	−10 dB to 15 dB

**Table 4 sensors-21-05319-t004:** Experimental parameters.

Parameters	Value	Parameters	Value
Transmit signal	FMCW	Target Number	0, 1 or 2
Carrier Frequency $f_{c}$	77 GHz	Target Range	10 m
Frequency Slope B	4.2 THz/s	Range Augmentation	9.7 m to 10.3 m
Bandwidth	3.5 GHz	Target Angle	−10° to 10°
Tx/Rx Antenna	2/4	Samples per chirp	512

## Data Availability

Training and testing data sets for the neural network models are available at: https://drive.google.com/drive/u/1/folders/1JGfJP9RrxgR1IozXMxpdOmnaoUaq_vdC (accessed on 6 August 2021).

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
