# Peer review of "DoA Estimation for FMCW Radar by 3D-CNN"

_sensors, 2021, doi:10.3390/s21165319_

Round 1

Reviewer 1 Report

The introduction should provide more details about the results of the proposed method. Additionally, the introduction should better describe how the proposed method outperforms the compared methods.

The authors describe the ESPRIT method as one of those used to compare the proposed approach. However, the article does not describe the ESPIRIT method. If the ESPIRIT method was in fact used, it should be described in the paper. If the ESPIRIT method was not used, the paper should be revised to remove it.

Please see the attached file for more details.

Reviewer 2 Report

Review on DoA Estimation for FMCW Radar by 3D-CNN

By Tzu-Hsien Sang *, Feng-Tsun Chien, Chia-Chih Chang, and Kuan-Yu Tseng, Bo-Sheng Wang, Jiun-In Guo

 This paper introduces a DoA estimation method for FMCW radars using 3D convolutional neural network. It is supposed to solve the problems of limited antenna number, low signal SNR, and unknown target number. The authors have explained the whole process and both simulation tests and field tests are performed. It is interesting to researchers who are trying to combine the DOA estimation with the deep learning methods. Still, I have the following concerns to be discussed:

  1. In the introduction part, the authors have explained many problems in the existing radar DOA estimation methods, such as low angle resolution, small measurement snapshots, and low signal SNR. But the most important improvement in DOA estimation after using the 3D-CNN is not clear. The authors should be specific on one or two features which can be improved by using 3d-CNN and also explain the reason behind.
  2. In page 2 line 50, the authors said ‘no convincing experiment results have been shown to support the claims’. The problem is, since there are no convincing experiment results, how do you know ‘Neural networks (NNs) have been shown more effective than MUSIC for estimating DoAs’ ?
  3. In page 5 line 153, the 3d-CNN is utilized as a DOA classifier with 21 classes. What if the DOA is higher than 10° or lower than -10°? Why do we set each class corresponds to DoAs −10°,−9°, ⋯ , 10° instead of −10.5°,−9.5°, ⋯ , 9.5°? The classification needs better explanation.
  4. About the experiment, what is the sampling frequency of the sampling device? How do you record signals with 77 GHz frequency?
  5. In Fig.5, it seems only two antennas are utilized. How did the authors constitute the virtual 8-antenna array? Would this affect the DOA estimation performance?
  6. Viewing from the results section, the authors have verified the better performance of 3d-CNN in DOA estimation accuracy. Thus, the improved estimation accuracy should be emphasized in the introduction section.

Reviewer 3 Report

The authors presented a method of direction-of-arrival (DoA) estimation for FMCW radar.  The introduction provides sufficient background and includes relevant references. The research design is appropriate. The methods are adequately described. The results are clearly presented. The conclusions are supported by the results. So, I recommend the acceptance of the paper with minor revisions for moderate English changes.
